# Mesoscopic calcium imaging in a head-unrestrained male non-human primate using a lensless microscope

Jimin Wu [1,7], Yuzhi Chen[2,3,7], Ashok Veeraraghavan [4,5], Eyal Seidemann [2,3] ✉ & Jacob T. Robinson [1,4,6] ✉

Mesoscopic calcium imaging enables studies of cell-type specific neural activity over large areas. A growing body of literature suggests that neural activity can be different when animals are free to move compared to when they are restrained. Unfortunately, existing systems for imaging calcium dynamics over large areas in non-human primates (NHPs) are table-top devices that require restraint of the animal's head. Here, we demonstrate an imaging device capable of imaging mesoscale calcium activity in a head-unrestrained male non-human primate. We successfully miniaturize our system by replacing lenses with an optical mask and computational algorithms. The resulting lensless microscope can fit comfortably on an NHP, allowing its head to move freely while imaging. We are able to measure orientation columns maps over a 20 mm² field-of-view in a head-unrestrained macaque. Our work establishes mesoscopic imaging using a lensless microscope as a powerful approach for studying neural activity under more naturalistic conditions.

Mesoscopic imaging provides an opportunity to discover how behaviors are represented by large-scale brain activity patterns that occur over multiple brain areas. The development and application of mesoscopic imaging effectively balances the need for high spatiotemporal resolution recording with remarkably large fields-of-view (FOVs). This technology has been demonstrated through its application in multiple species including rodents[1–4], cats[5–9] and primates[10–15]. When applied to rhesus macaques, mesoscale imaging can help uncover cognitively sophisticated behaviors like perception, motor planning and decision-making because these non-human primates (NHPs) have close genetic, anatomical and behavioral similarity to humans[16–18]. Imaging provides unique advantages for researchers to *non-invasively* record high density neural activity from large neuron populations[13,19–21]. For example, one can study perception in the primary visual cortex (V1). Specifically, primates display pinwheel-like structures that segregate neurons into columns based on edge orientation with periodicity of ~1.2 cycles/

mm[15,22–25], while rodents have a random distribution of orientation-selective neurons, resulting in salt-and-pepper organization[24,26]. The success of these efforts can be attributed to the refinement of surgical procedure, and improvement of genetically encoded calcium indicators (GCaMP)[27,28] and the associated viral preparation[13,29–35]. The use of transparent windows implanted over the brain enables large-area, non-invasive optical access to the brain, allowing researchers to study neural activity in a more comprehensive manner. However, these studies have been mostly conducted on head-fixed macaques which is unnatural. Virtual reality environments have been developed to address this challenge, but these environments lack vestibular input and disrupt eye-head movement coupling[36].

Recent advancements in miniaturized head-mounted imaging devices or "miniscopes" enabled the study of neural activity in freely-moving rodent models, but are typically restricted to small FOVs less than 1 mm[2,37–40]. While there are ongoing efforts to develop

[1]Department of Bioengineering, Rice University, 6100 Main Street, Houston, TX 77005, USA. [2]Department of Neuroscience, University of Texas at Austin, 100 E 24th St., Austin, TX 78712, USA. [3]Department of Psychology, University of Texas at Austin, 108 E Dean Keeton St., Austin, TX 78712, USA. [4]Department of Electrical and Computer Engineering, Rice University, 6100 Main Street, Houston, TX 77005, USA. [5]Department of Computer Science, Rice University, 6100 Main Street, Houston, TX 77005, USA. [6]Department of Neuroscience, Baylor College of Medicine, One Baylor Plaza, Houston, TX 77030, USA. [7]These authors contributed equally: Jimin Wu, Yuzhi Chen. ✉e-mail: eyal@austin.utexas.edu; jtrobinson@rice.edu

head-mounted devices that can perform mesoscale calcium imaging over large FOVs, the resolution of previously reported mesoscopes is around 40–50 μm[41], which may struggle to resolve sub-columnar scale features. Some recent studies have utilized miniature, integrated one-photon fluorescence microscopes to image naturally behaving NHPs, and recorded head-mounted calcium dynamics from primary motor cortex of marmosets[42] and dorsal premotor cortex of macaques[43]. These studies represent important progress in understanding neural circuit mechanisms in NHPs under more naturalistic conditions. However, these small FOVs limit our ability to study neural activity across multiple brain regions or at the scale of multiple cortical columns. In the case of the primary visual cortex (V1) of macaques, the spatial scale of the retinotopic map and orientation map spans multiple mm[2,12,25,44–47], which cannot be observed with the small FOVs provided by the current miniscopes. A recent study[48] demonstrated a head-mounted device for macaques capable of doing intrinsic imaging over a cm² FOV with ~100 μm spatial resolution at 5 Hz acquisition. This promising new approach enables imaging during more naturalistic behaviors, but it has yet to demonstrate fluorescence imaging or achieve the sub-100 um resolution necessary to accurately resolve sub-columnar scale features. A miniaturized, head-mounted device with a large FOV and high resolution for imaging NHPs has the potential to enhance our understanding of mesoscale cortical dynamics under more naturalistic conditions.

Current lens-based optical techniques for measuring neural activity are limited by a fundamental tradeoff between FOV, resolution and the size of the device. In this study, we present a miniaturized lensless microscope that can efficiently address this tradeoff. In lens-based systems, the scene is projected directly onto the imaging sensor with magnification, while lensless imaging systems produce an invertible transformation of the scene[49–56]. The captured images are usually highly multiplexed but can be reconstructed using reconstruction algorithms[51,57]. By removing lenses and incorporating a 'contour' phase mask[49,58], our system demonstrates substantial improvements in FOV, allows digital refocusing, and achieves a compact, lightweight form factor. With a weight of 17.2 g, our system can be comfortably worn on the head of the NHP, allowing for freedom of movement during the imaging. We refer to this prototype as the 'Bio-FlatScopeNHP', because it uses a contour phase mask similar to the mask used for the previously reported 'Bio-FlatScope'[49], with the addition of key hardware and algorithmic innovations that enable large FOV imaging in a behaving NHP. Specifically, we developed a reconstruction model that considers the spatial variance of the point spread functions (PSFs), leading to enhanced resolution and a wider FOV. Moreover, we demonstrated the integrated illumination system specifically designed for head-mounted in vivo imaging. These innovations allow the Bio-FlatScopeNHP to image with a resolution of better than 10 μm over a ~20 mm² FOV in vivo with integrated illumination, and a 64 mm² FOV without illumination constraints. This FOV is approximately 3 times larger than miniature microscopes that have a spatial resolution of 10 μm or better (Fig. 1f). Using this prototype, we performed functional imaging in V1 of a macaque with a genetically encoded calcium indicator GCaMP6f. We accurately measured the large-scale retinotopic map and a finer scale columnar orientation map of V1 on a head-fixed macaque, which showed very good correspondence to ground truth captured by a traditional table-top widefield microscope. More importantly, we have demonstrated the functional imaging in V1 of a head-unrestrained macaque, and successfully measured columnar-scale population responses in V1. Bio-FlatScopeNHP has a weight of only 17.2 g, which is lighter than head-mounted devices used in rats[59], ferrets[60] and common marmosets[61], suggesting that this microscope could be applied generally to other animals to study how brain function underlies complex behavior under naturalistic conditions (Supplementary Fig 1).

## Results

### System design and characterization

We designed Bio-FlatScopeNHP using a 'contour' phase mask[49,58] with a pattern width of 6 μm, a hybrid filter set[49,62,63], a monochromatic imaging sensor and an illumination system. As previously reported, successful lensless imaging in complex biological imaging requires a high-contrast and spatially localized point spread function (PSF), capable of capturing all directional spatial frequencies[49]. To create the PSF pattern, we generated Perlin noise and applied Canny edge detection[58] (Supplementary Fig 3). We then fabricated the phase mask using a two-photon lithography system (Nanoscribe, Photonic Professional GT). The illumination system consists of four blue light-emitting diodes (LEDs) paired with excitation filters, and housed in a 3D printed housing (Fig. 1b). The device weighs 8.6 g without the supporting holder and 17.2 g with the holder. This lightweight system can be easily mounted on the head of a macaque without compromising its natural behavior.

We first performed system calibration by using a point source to capture the PSFs at a working distance range of 0.5 mm–6 mm. By using PSFs at different working distances, the system can achieve high-quality imaging at a wide range of working distances. The distance from the device surface to the cortex chamber surface is approximately 3 mm, and the example images shown in Fig. 2 were all captured at the 3 mm working distance.

When performing resolution tests using a negative fluorescent 1951 USAF Resolution Target (Edmund Optics 59-204) we observed a < 10 μm resolution within 3.5 mm working distance (Fig. 2b). At each distance, we captured 5 images at 10 ms exposure time each and averaged them to reduce noise. Figure 2c shows an example reconstructed image taken by Bio-FlatScopeNHP at the distance we performed in vivo imaging, where lines in group 5 element 5 are clearly visible, indicating a resolution of 9.84 μm.

Using Bio-FlatScopeNHP, we reconstructed images of a slice of Convallaria Rhizome (lily of the valley) and a sample containing spiking human embryonic kidney (HEK) cells with resolution comparable to a 4x microscope objective (Fig. 2). This demonstrated our capability to accurately reconstruct low-contrast biological samples at a FOV of ~20 mm². We first imaged a Convallaria Rhizome slice stained with Acridine-Orange (BostonElectronics). The ground truth images were taken by using a 4x microscope objective (Nikon Fluor) and compared to the Bio-FlatScopeNHP reconstructions (Fig. 2d). We captured 5 images at 50 ms exposure time each and averaged for noise removal. The zoom-in of the reconstruction shows that Bio-FlatScopeNHP can clearly resolve large circular plant cell structures with a size of around 10 μm, and the reconstruction has a good correspondence to the ground truth image. Additionally, we imaged a sample containing spiking human embryonic kidney (HEK) cells stained with Calcein-AM[64]. The HEK cells were held on a photolithographically patterned 12 mm coverslip with polydimethylsiloxane for 300 μm circles every 300 μm[65]. We were able to image a FOV of ~20 mm² (Fig. 2e) using integrated illumination with the pattern clearly visible in the reconstruction. The zoom-ins (Fig. 2f) shows that our system can resolve single HEK cells with good correspondence to ground truth.

### In vivo calcium imaging of position tuning property in a head-fixed macaque

We found that Bio-FlatScopeNHP is capable of imaging large scale calcium dynamics on behaving macaques. When we examined the tuning property of V1 GCaMP response across different stimulus positions on a head-fixed behaving macaque using Bio-FlatScopeNHP, we found a good correspondence between the results captured by Bio-FlatScopeNHP and the ground truth captured by a table-top widefield microscope[13,25,66] (Fig. 3). Bio-FlatScopeNHP imaging was performed under the same conditions as the table-top widefield microscopic recording, within a two-hour time window.

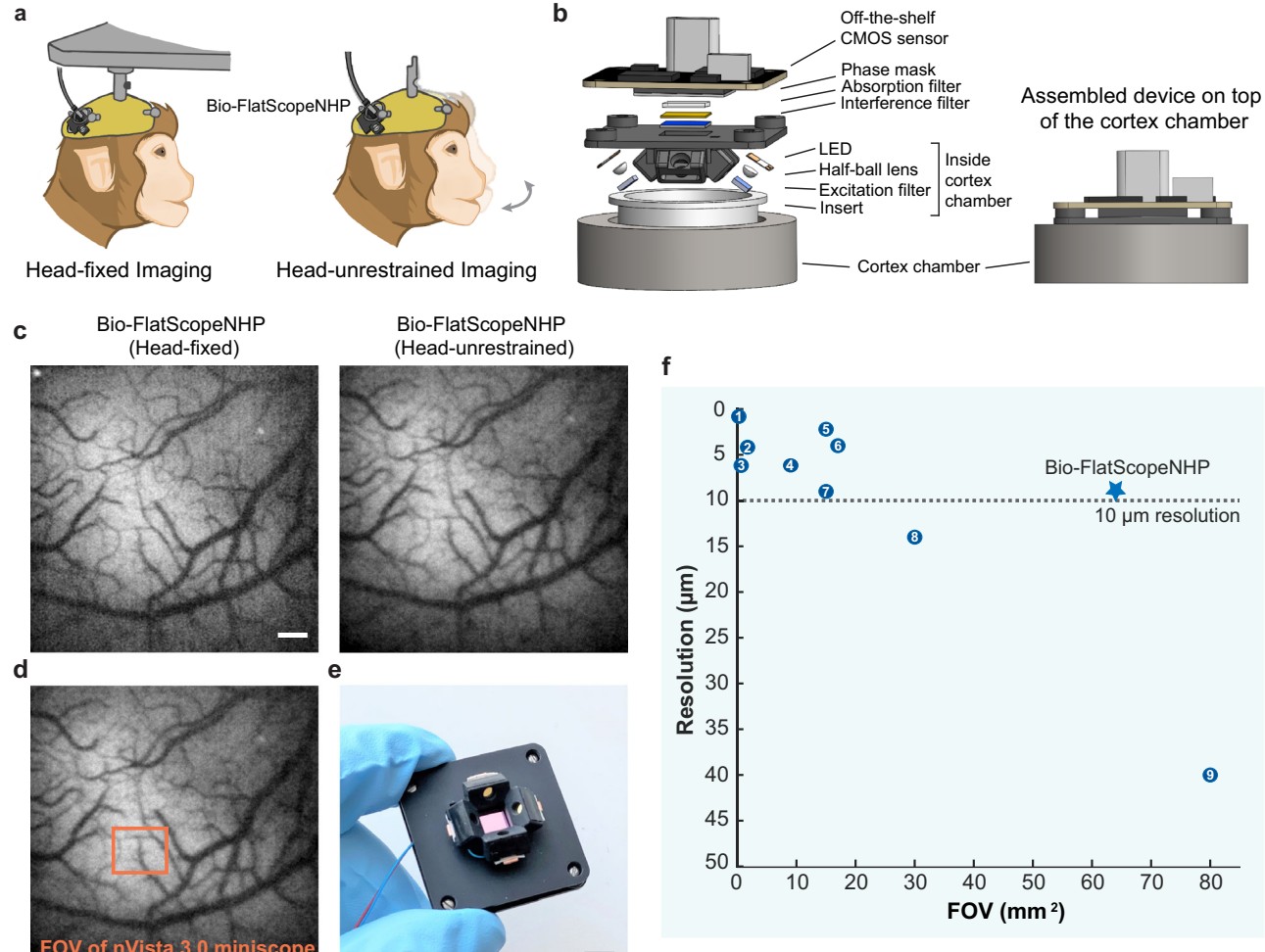

**Fig. 1 | A miniaturized lensless fluorescence microscope for in vivo imaging on head-unrestrained NHPs. a** Illustration of experimental setups for imaging a head-fixed animal (left) and head-unrestrained animal (right) using Bio-FlatScopeNHP. **b** CAD rendering of the Bio-FlatScopeNHP with integrated illumination system. The device incorporates a fabricated phase mask, a hybrid filter set to filter out excitation light, an off-the-shelf CMOS sensor and the illumination from four blue LEDs. The integrated illumination module goes into the cortex chamber when in vivo imaging, with a ~3 mm working distance (distance between the sample and the bottom of the device). The sensor to sample distance is ~12 mm. **c** In vivo fluorescence images reconstructed by Bio-FlatScopeNHP captures on a head-fixed NHP (left) and a head-unrestrained NHP (right). Scale bar, 500 μm. **d** A field-of-view

(FOV) comparison with current technology used for imaging calcium activities on a head-unrestrained NHP[43]. **e** A photo of an assembled Bio-FlatScopeNHP prototype. Scale bar, 5 mm. **f** Comparison of Bio-FlatScopeNHP to head-mounted fluorescence microscopes implemented in rodent models. Star indicates the resolution and achievable FOV of Bio-FlatScopeNHP. Data points (1-9) in the figure: 1. FHIRM-TPM[37]; 2. Miniscope V4[77]; 3. MiniLFM[38]; 4. miniscope-LFOV[78]; 5. Mesoscope[79]; 6. Kiloscope[80]; 7. Bio-FlatScope[49]; 8. cScope[59]; 9. mini-mScope[41]. The resolution reported here is the peak resolution at the center of the FOV of each device, but it should be noted that the resolution near the edges of these microscopes is less than that at the center.

For these experiments, the animal was implanted with a metal head post and a recording chamber over the dorsal portion of V1, and a transparent artificial dura was used to provide optical access and protect the brain surface[13] (Supplementary Fig 5). In our experiments, GCaMP6f was used as the reporter, and excitatory neurons were targeted with GaMKIIa promotor[13]. The imaging experiments were performed around 1.5 years post injection.

Macaque V1 has been widely studied and considered as one of the best characterized cortical regions in the primate brain. One important feature of macaque V1 is its topographic organization, which encompasses a large-scale retinotopic map representing the contralateral visual field[46,67,68] as well as a finer-scale orientation map[7,44,66]. We first focused on the retinotopic map. When performing the experiments, we selected a visual stimulus pattern that has been reported to elicit reliable position-tuning curves in macaque V1[13,25,66]. Specifically, we used a small flashed grating as the visual stimulus with 0.25 degree stimulus size and 4 cycle per degree (cpd) spatial frequency presented at different locations in the visual field. Stimulus position vertical

coordinates in degree of visual angle varied from −0.8 degree to −1.3 degree, with a change of 0.1 degree change (Fig. 3c). During the experiments, the macaque performed a visual fixation task, keeping fixation within a window of less than 2 degree width, centered on a small fixation point for 2–4 s. An infrared eye tracker (EyeLink) was used to monitor the animal's eye position. A second identical grating was flashed at a mirror-symmetric location in the opposite hemifield to help the macaque maintain eye fixation. The stimuli were flashed 4 times per trial at 4 Hz frequency (100 ms ON and 150 ms OFF). All stimulus conditions were randomly interleaved and repeated for 10 times each, and the stimulus trials were mixed with blank fixation trials. The recorded videos were captured at 20 Hz for both Bio-FlatScopeNHP and widefield microscope.

From the Bio-FlatScopeNHP reconstructions, it is clear that GCaMP signals were entrained with each stimulus cycle suggesting that we were indeed recording neural activity with the Bio-FlatScopeNHP (Fig. 3b). We first registered the Bio-FlatScopeNHP reconstructions with those from widefield microscopy based on blood

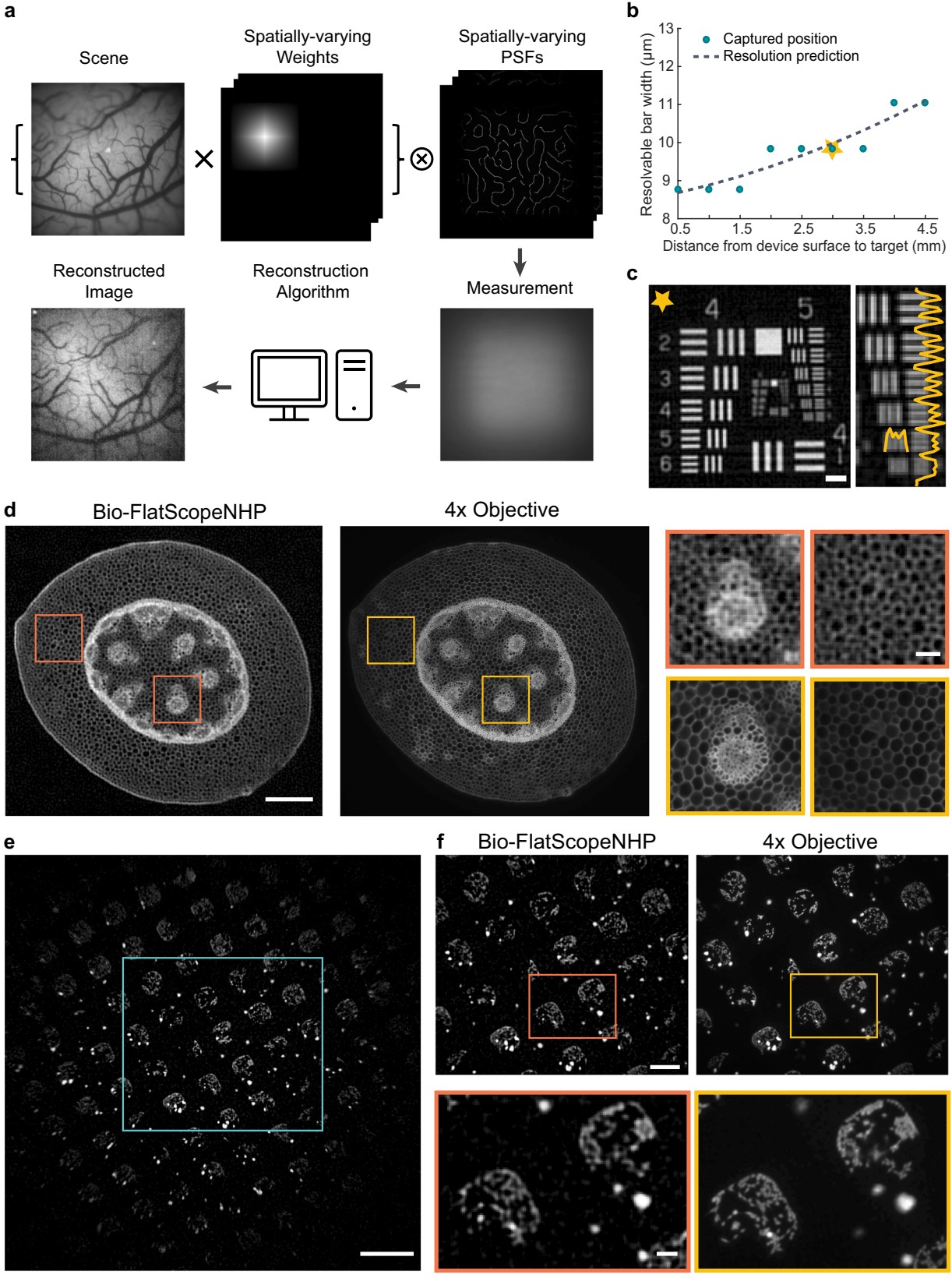

vessel structures. From the reconstructions of the response to 6 stimuli at different positions, it is evident that the active areas captured by both devices show good correspondence (Fig. 3f). We selected six regions of interest (ROIs) for tuning curves analysis, with a size of ~0.09 mm² each and the distance of around 0.37 mm between adjacent ROIs (Fig. 3d, e). We recorded 2 sessions of Bio-FlatScopeNHP

experiments (additional results in Supplementary Fig 10) and 1 session of a widefield microscope experiment. The Gaussian fitted tuning curve for the 6 ROIs shows excellent agreement between Bio-FlatScopeNHP reconstructions and widefield microscope captures (Fig. 3d, e). The peak position of each fitted tuning curve indicates the corresponding visual location at each ROI, which shows a difference

**Fig. 2 | High resolution images of fixed fluorescent samples compare with ground truth. a** Overview of Bio-FlatScopeNHP imaging procedure. PSFs: point spread functions. **b** System resolution tested using a USAF resolution target at different imaging depths. Yellow star indicates the actual imaging depth for in vivo imaging. **c** A representative USAF target reconstruction at the actual imaging depth for in vivo imaging. Scale bar, 100 μm. **d** High resolution images of a stained slice of Convallaria Rhizome captured by Bio-FlatScopeNHP and the ground truth 4X objective. Scale bars: whole Convallaria Rhizome, 500 μm; zoom-ins, 20 μm. **e** Patterned live spiking HEK293 cells imaged by Bio-FlatScopeNHP with integrated illumination. Scale bar, 1 mm. **f** Zoom-in of the blue square area in panel f of Bio-FlatScopeNHP which matched the FOV of the ground truth 4X objective. Scale bar, 500 μm. Zoom-ins below show that Bio-FlatScopeNHP is able to resolve HEK cell structures inside the square pattern. Scale bar, 100 μm.

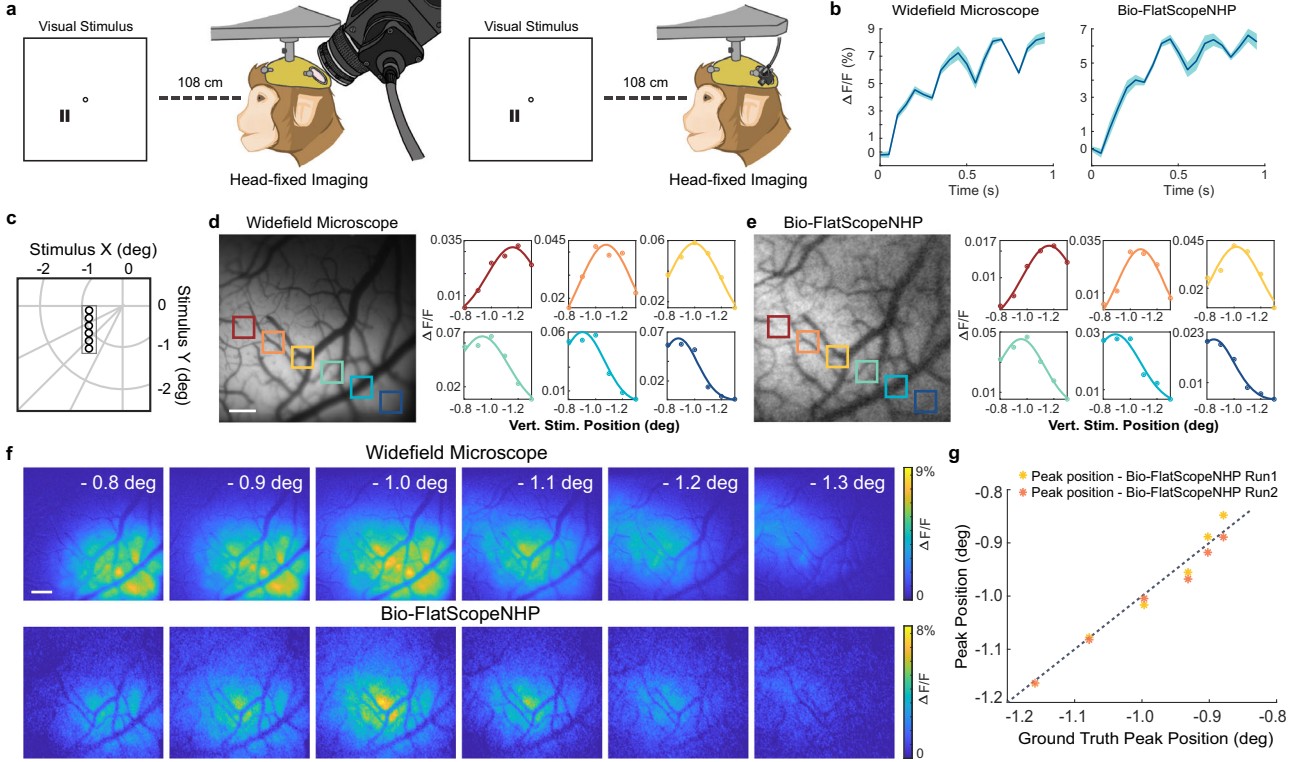

**Fig. 3 | Spatial response profiles measured from V1 in the right hemisphere of a head-fixed macaque in one GCaMP imaging experiment. a** Illustration of experimental setups for capturing ground truth (left) and Bio-FlatScopeNHP imaging (right). Small gratings at 6 different positions are used as visual stimuli in the experiments. **b** Average time course of GCaMP response to the flashed gratings over the center 2 mm × 2 mm area of the FOV from ground truth (left) and Bio-FlatScopeNHP reconstructions (right). ΔF/F indicates the relative changes in fluorescence. Shaded area ± SEM. **c** Visual stimulus coordinates in degrees of visual angle. **d** An example image captured by ground truth widefield microscope (left). Scale bar, 500 μm. Spatial distribution of response amplitudes for different stimulus positions fitted by a Gaussian function in ground truth captures (right). The data points shown in the right panel represent the mean response measured within the cortical regions highlighted by color coding in the left panel. The main response from blank trials is subtracted correspondingly. **e** An example image captured and reconstructed using Bio-FlatScopeNHP (left) and fitted spatial distribution (right) from the reconstructions. **f** Response amplitude of ground truth and Bio-FlatScopeNHP at different stimulus conditions. Scale bar, 500 μm. **g** Peak position comparison between ground truth captures and Bio-FlatScopeNHP reconstructions. Source data are provided as a Source Data file.

between two devices of less than 0.031 degree (average difference = 0.014 degree) (Fig. 3g).

Despite the high level of agreement between Bio-FlatScopeNHP reconstruction and widefield microscope captures, we noticed a lower ΔF/F in Bio-FlatScopeNHP reconstructions compared to the widefield microscope, especially at the edge of the FOV. One possible explanation could be the lower signal-to-noise ratio in Bio-FlatScopeNHP reconstructions, and the illumination profile can also potentially affect the reconstruction quality (Supplementary Fig 7). Nevertheless, this experiment confirmed that Bio-FlatScopeNHP can capture calcium dynamics and obtain high-quality position-tuning information that is comparable to traditional table-top widefield microscopes.

## In vivo calcium imaging of orientation tuning maps in a head-fixed macaque V1

We further found that Bio-FlatScopeNHP is capable of capturing calcium dynamics at a columnar scale comparable to a table-top widefield

microscope system. Obtaining high-quality orientation maps is more challenging than position-tuning curves due to the relatively small spatial scale and weak orientation-selective signals.

In this experiment, we used grating visual stimuli that have been previously reported to reliably elicit orientation selective signals in macaque V1[12,13,66]. While the macaque maintained a steady gaze at the center of the screen, large high contrast sinusoidal grating stimuli at 6 equally spaced orientations (0, 30, 60, 90, 120, 150 degrees) were flashed at a frequency of 4 Hz for 4 cycles, with a spatial frequency of 4 cpd. A second identical grating was flashed at a mirror-symmetric location in the opposite hemifield to help the macaque maintain eye fixation. The size of the grating is 6 deg x 6 deg, which is large enough to cover the whole imaging area. All stimulus conditions were randomly interleaved, repeated for 10 times each, and mixed with blank fixation trials. We recorded 2 sessions of Bio-FlatScopeNHP experiments (additional results in Supplementary Fig. 11) and 2 sessions of widefield microscope experiments.

We obtained precise orientation maps from the Bio-FlatScopeNHP reconstructions and observed strong GCaMP signals during each stimulus cycle (Fig. 4b) with ΔF/F levels comparable to those captured by the widefield microscope. We first registered the Bio-FlatScopeNHP reconstructions with those from the widefield microscope using blood vessel structures as a reference. Using root-mean-square (RMS) maps calculated on averaged amplitude of the orientation tuning at each pixel location (Supplementary Fig 9), we selected ROIs with strongest 4 Hz signals (Fig. 4c). To analyze the orientation maps from the captured signals, we computed the 4 Hz Fourier amplitude of the average GCaMP signal at each location, and applied a bandpass spatial filter (0.8–2.5 cycle/mm) to remove non-orientation-selective responses and high-frequency noise (Fig. 4f). We employed a linear orientation decoder[69] to quantitatively evaluate the orientation discriminability of the captured columnar-scale neural responses, as follows. We first calculated pixel-wise d' maps from the set of single-trial 0 degree and 90 degree orientation maps,

where each pixel showed the d' for discriminating between the two orientations at that pixel. These d' maps were then used as linear weights to compute a decision variable per trial for each orientation (Fig. 4e). The separability of these decision variables between the two orientations was used to compute an overall d' (see Methods). The decoder uses pixel-wise summation of single-trial responses, weighted by corresponding d' maps, and summed over the selected ROI (see Method). With cross-validation (see Method), we calculated the discriminability of the pooled signals in distinguishing between 0 degree and 90 degree orientations, which can be interpreted as the signal-to-noise ratio of the decoder. We found a high discriminability (d') in both the ground truth captures (d' = 29.90) and Bio-FlatScopeNHP reconstructions (d' = 6.58) (Fig. 4e). Representative maps of 0 degree and 90 degree are shown in Fig. 4d. To verify the validity of the observed map, we computed the Pearson correlation between each pair of maps, and averaged the correlations across all pairs of maps with the same stimulus orientation

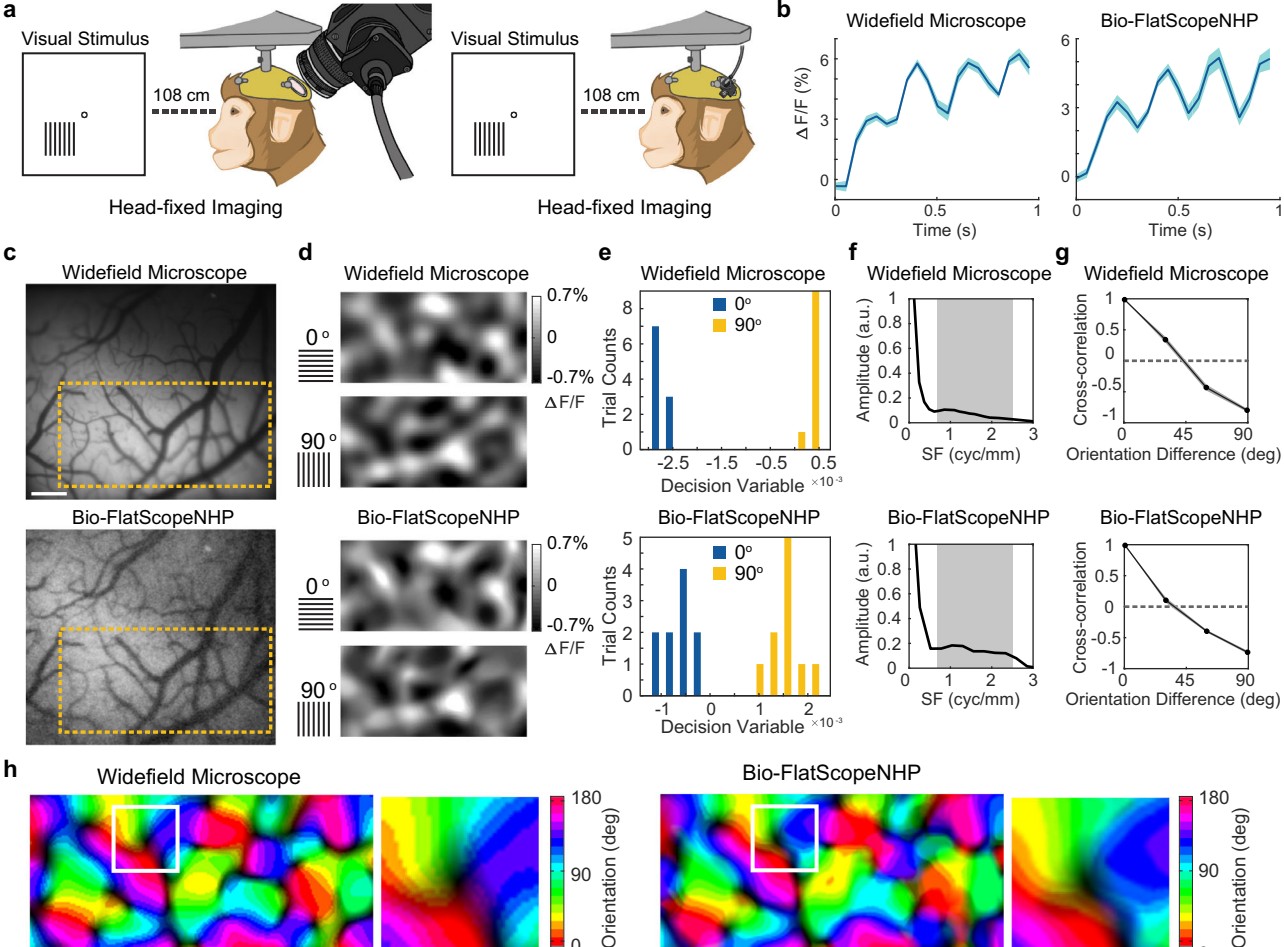

**Fig. 4 | Columnar-scale orientation tuning maps measured from V1 in the right hemisphere of a head-fixed macaque in one GCaMP imaging experiment.** **a** Illustration of experimental setups for capturing ground truth (left) and Bio-FlatScopeNHP imaging (right). In the experiments, flashed large gratings at six different orientations are used as visual stimuli. **b** Average time course of GCaMP response to the flashed gratings from ground truth (top) and Bio-FlatScopeNHP reconstructions (bottom). ΔF/F indicates the relative changes in fluorescence. Shaded area ± SEM. **c** Left: Example images captured by ground truth widefield microscope (top) and Bio-FlatScopeNHP (bottom). Scale bar, 500 μm. The yellow square indicates the selected overlapping ROI. **d** Response maps of orientation-selective GCaMP signals obtained by bandpass filtering the response maps to 0-degree and 90-degree orientations (out of six evenly spaced orientations)

obtained by ground truth widefield microscope (top) and Bio-FlatScopeNHP (bottom). **e** Decision variables calculated from 0-degree and 90-degree trials captured by ground truth widefield microscope (top) and Bio-FlatScopeNHP (bottom). **f** Spatial 1D amplitude spectrum of the single-orientation response maps obtained by ground truth widefield microscope (top) and Bio-FlatScopeNHP (bottom). The spatial filtration removes components outside spatial frequency (SF) between 0.8 and 2.5 cycles/mm, indicated by the shaded area. **g** Pairwise correlations between all six orientation maps as a function of stimulus orientation difference obtained by ground truth widefield microscope (top) and Bio-FlatScopeNHP (bottom). Shaded area ± SEM. **h** Orientation maps obtained by ground truth widefield microscope (top) and Bio-FlatScopeNHP (bottom). Scale bar, 500 μm. Source data are provided as a Source Data file.

difference (Fig. 4g). As the difference in orientation between two maps increases, their correlations follow a systematic pattern. When the two orientations differ by 45 degrees, the correlation approaches zero, and for an orientation difference of 90 degrees, the correlation reaches a maximum negative value of −0.81 for ground truth and −0.75 for Bio-FlatScopeNHP, which matched the known structure of orientation maps in V1[12,13,66]. We then used the maps at each orientation to compute the composite orientation map (Fig. 4h), in which the color indicates the preferred orientation and saturation the strength of orientation tuning. This composite map reveals the semi-periodic organization of the orientation map, including the orientation pinwheels[7,69]. To assess the accuracy of Bio-FlatScopeNHP captures, we compared the composite orientation maps obtained from Bio-FlatScopeNHP and the widefield microscope by calculating the correlation coefficient. We converted the color map of preferred orientation to grayscale map in the range of [−1 1] by taking the sine of the preferred orientation times two[13]. We computed the correlation coefficients between the converted maps in each pixel from the maps calculated from Bio-FlatScopeNHP and those obtained using the widefield microscope. The correlation coefficients between the maps calculated from Bio-FlatScopeNHP and the widefield microscope were found to be similar (correlation coefficients = 0.87 and 0.85) as compared to the correlation coefficient between two sessions captured by the widefield microscope (correlation coefficient = 0.92). This experiment demonstrated that Bio-FlatScopeNHP is capable of capturing calcium dynamics at columnar scale and producing high quality orientation map information that is comparable to traditional table-top widefield microscopes.

## In vivo calcium imaging of orientation tuning maps in a head-unrestrained macaque V1

Since the Bio-FlatScopeNHP produces images of similar quality to those captured by the table-top microscope system for head-fixed macaques, we were able to release the head fixation and achieve an important milestone - imaging columnar-scale neural activity in a head-unrestrained macaque.

By imaging a head-unrestrained macaque, we demonstrated that Bio-FlatScopeNHP can successfully capture calcium dynamics at a columnar scale even when the animal's head is unrestrained. The most significant advantage of Bio-FlatScopeNHP compared to traditional table-top systems is its small form factor, which allows it to be mounted directly on the intracranial implant. In this experiment, we took advantage of this feature and imaged the orientation tuning properties of V1 GCaMP response across different stimulus orientations on a head-unrestrained macaque. In the same experimental session, we also recorded a head-fixed block of trial using the widefield microscope and a head-fixed Bio-FlatScopeNHP block (Supplementary Fig 12) for comparison, all within a two-hour time window. Same visual stimulus with 6 different orientations was used in the widefield microscopic session. For the head-unrestrained experiment, we moved the visual stimulus to half distance (54 cm) and enlarged the visual stimulus pattern to 20 degree stimulus size and 3 cpd spatial frequency presented at different orientations (Fig. 5a). Enlarging the stimulus size alleviates the requirement for strict eye fixation under head-unrestrained conditions. The subject was trained to look at the fixation point at the center of the screen after an audio cue to get a reward. However, in contrast to the head-fixed condition, we did not enforce fixation in the head-unrestrained trials.

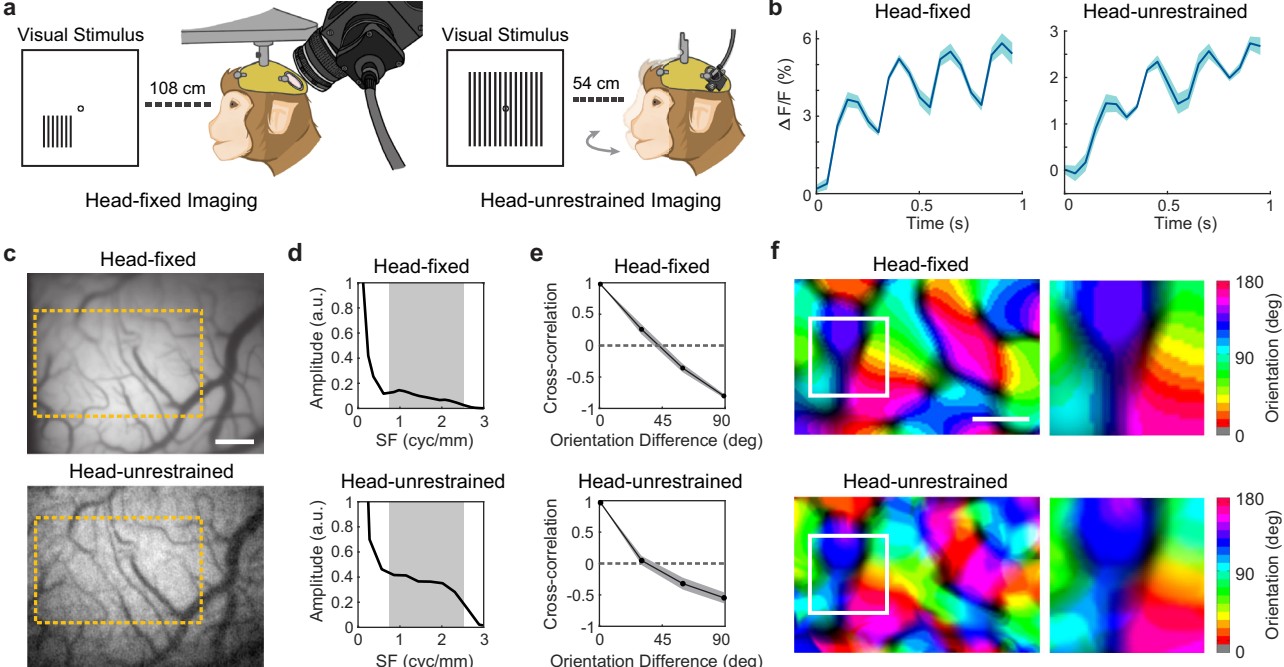

**Fig. 5 | Columnar-scale signals measured from V1 in the right hemisphere of a head-unrestrained macaque in one GCaMP imaging experiment. a** Illustration of experimental setups for head-fixed imaging (left) and head-unrestrained imaging (right). In the experiments, large gratings at six different orientations are used as visual stimuli. During head-unrestrained experiments, visual stimuli were moved closer to the animal and the stimulus was larger and at a higher spatial frequency. **b** Average time course of GCaMP response to the flashed gratings from head-fixed imaging (left) and head-unrestrained imaging (right). ΔF/F indicates the relative changes in fluorescence. Shaded area ± SEM. **c** Example images captured by widefield microscope (top) in a head-fixed imaging session and Bio-FlatScopeNHP (bottom) in a head-unrestrained session. Scale bar, 500 μm. The yellow square indicates the selected overlapping ROI. **d** Spatial 1D amplitude spectrum of the single-orientation response maps obtained by head-fixed imaging (top) and head-unrestrained imaging (bottom). The spatial filtration removes components outside spatial frequency (SF) between 0.8 and 2.5 cycles/mm, indicated by the shaded area. **e** Pairwise correlations between all six orientation maps as a function of stimulus orientation difference obtained by head-fixed imaging (top) and head-unrestrained imaging (bottom). Shaded area ± SEM. **f** Orientation maps obtained by head-fixed imaging (top) and head-unrestrained imaging (bottom). Scale bar, 500 μm. Source data are provided as a Source Data file.

We were able to obtain an accurate orientation map from the Bio-FlatScopeNHP reconstructions on the head-unrestrained macaque and observed clear GCaMP signals during each stimulus cycle (Fig. 5c) with a decreased ΔF/F compared to the head-fixed captures. We registered the reconstructions from head-unrestrained sessions with those from the widefield microscope captures based on blood vessel structures. We used the same RMS maps to select the region of interest as in previous experiments, and applied the same bandpass spatial filter (0.8–2.5 cycle/mm) on the 4 Hz Fourier amplitude of the averaged GCaMP signal to remove non-orientation-selective responses and high frequency noise (Fig. 4d). To verify that observed maps are actual orientation columns, we computed the Pearson correlation between each pair of maps, and averaged the correlations across all pairs of maps with the same stimulus orientation difference (Fig. 5e). The pairwise correlations obtained by Bio-FlatScopeNHP from the head-unrestrained animal still smoothly change from positive values for maps produced by nearby orientations to negative values for maps produced by orthogonal orientations, reaching a maximum negative value of −0.81 for head-fixed session and −0.54 for head-unrestrained session (Fig. 5e). We calculated the map captured from Bio-FlatScopeNHP imaging of the head-unrestrained animal, and compared it to the map obtained from a head-fixed animal using the widefield microscope (Fig. 5f). The map from the head-unrestrained animal still revealed the semi-periodic organization of the orientation map, and the orientation pinwheels[44] are still visible but with blurrier edges. To evaluate how closely the head-unrestrained orientation maps matched the head-fixed condition, we computed the correlation to random orientation maps by randomly labeling the 6 orientations from the head-unrestrained animal 1500 times and calculating the correlation coefficient. The maps generated from the shuffled data showed random correlation coefficients ranging from −0.08 to 0.48 (average = 0.25). The actual correlation coefficient between the head-unrestrained animal and the head-fixed animal is 0.72, indicating that these maps show significant similarities that are not explained by chance (Supplementary Fig. 14).

## Discussion

In this study, we present a lensless microscope capable of imaging large FOVs (over 20 mm²) at high resolution in the primary visual cortex of the macaque. The small and light form factor allows us to perform widefield in vivo calcium imaging on a head-fixed awake macaque from which we were able to extract position tuning properties and orientation maps. These maps showed good correspondence with those captured by a traditional table-top widefield microscope. A key advantage of head-unrestrained imaging is the ability to study eye-head movement coordination during natural behaviors. During natural behaviors, the head and eyes move simultaneously and in a coordinated fashion. Because it has been previously impossible to imaging large-scale neural activity during this complex behavior, many open questions remain regarding the neural mechanisms that mediate the coordination of the head and eyes and the coordinate transformations that are necessary to guide motor responses based on multimodal sensory evidence. Our technology would allow researchers to study how sensory areas, such as V1 and auditory cortex[70,71], and sensory motor areas[72,73], such as the premotor cortex and frontal eye fields, are affected by natural head and eye movements. Another area where the Bio-FlatScopeNHP could play an important role is understanding social interactions[74,75] and emotions[76] in primates. Macaques are highly social animals and recent studies have started to look at the neural basis of social interactions between NHPs. To better understand social behavior and its underlying neural representations, it is essential to study behavior and neural representations in head unrestrained animals. While many of these questions may be addressable with head-unrestrained animals, some of these questions would be best studied in freely behaving animals. To facilitate studies in freely behaving

animals, future versions of the Bio-FlatScopeNHP should be made wireless, protected in a more robust housing, and more robust hemodynamic correction should be added. In this proof-of-principle study in a head-unrestrained animal, we used heartbeat-triggered trials and a fast, periodic stimulus presentation to overcome hemodynamic artifacts and improve SNR. For future continuous recording under more naturalistic conditions, one could compensate for the hemodynamic artifacts by recording the reflectance signal using a color that does not interfere with the fluorescence channel, as has been used in rodent imaging. For example, GCaMP imaging in mice can achieve hemodynamic correction by using a color camera to record the reflectance signal of green and red LEDs[2,4,41]. Future versions of the Bio-FlatScopeNHP could use a similar approach by changing the emission filter. Alternatively, future versions of the Bio-FlatScope NHP could use interleaved blue and green illumination with a trigger circuit to image fluorescence and reflectance signal simultaneously[41], which has been shown in rodent imaging.

One potential limitation of our approach is that the overall signal-to-noise ratio (SNR) in Bio-FlatScopeNHP reconstructions is lower compared to traditional table-top systems, particularly at the edge of the FOV. This can lead to a decrease in ΔF/F compared to the widefield microscope imaging. This could be attributed to uneven illumination at the edges of the FOV. However, this issue can be improved by optimizing the phase mask and illumination setups to allow imaging at closer distance at better illumination conditions. Imaging at closer distance can increase the light collection efficiency and improve the system resolution and SNR. Another contributing factor is that the imaging experiments in this study were conducted approximately 1.5 years post injection, which could have led to a reduction in signal levels. When imaging is performed on a subject that has recently received the injection, it is likely that we could achieve higher signal levels and an improved SNR.

An additional constraint limiting the size of our in vivo FOV is the dimensions of the chamber insert. Our illumination design is restricted by the small size (21 mm diameter) and the large depth (8 mm) of the chamber insert. When removing this constraint, our device demonstrates the capability to achieve a significantly expanded FOV (64 mm²), as shown in Supplementary Fig 4 and Supplementary Fig 7d. Compared to current head-mounted devices used in rodents[37,38,41,49,59,77–80], the Bio-FlatScopeNHP stands out as the only system that combines a peak resolution of <10 µm and a FOV over 60 mm² (Fig. 1f). Although our system demonstrated a <10 µm resolution on test target, individual fluorescent cells in vitro, and NHP blood vessels roughly 20 µm in diameter (Supplementary Fig 5e), the dense GCaMP labeling and scattering by the brain tissue prevents single-cell resolution fluorescent imaging in vivo. This is also true for wide-field imaging with similar dense labeling, where individual cells cannot be resolved. Given the <10 µm optical resolution of the Bio-FlatScopeNHP we expect that single-cell resolution imaging may be possible when sparse labeling strategies developed in mice[81] are translated to NHPs. These strategies include using cell-type specific promoters to label sparse neural populations such as subclasses of inhibitory neurons[82,83], combining retrograde AAV injections in downstream regions[84,85] together with intersectional approaches such as *Cre* or *Flp*[82] to strongly label sparse populations of projections cells from superficial layers, and using genetic motifs that restrict GECI expression to the soma.

While we demonstrate the Bio-FlatScopeNHP in rhesus macaque, the lightweight design should be compatible with a number of other common animal models. The largest contributor to the weight of our current prototype (weight ~17.2 g) is our off-the-shelf CMOS sensor (~4.9 g) and 3D printed housing (~8.6 g) used for mounting on the cortex chamber. We compared the weight of the Bio-FlatScopeNHP with head-mounted devices currently utilized in several other animal models, including rats[59], ferrets[60] and common marmosets[61]

(Supplementary Fig 1). Notably, the overall weight of the Bio-FlatScopeNHP is lighter than the weight that can be carried on these animals, which indicates the potential for in vivo imaging studies across species. We envision our device as a promising candidate for conducting in vivo imaging with large FOVs on marmosets - an emerging NHP model system in neuroscience. Marmosets have shallower cortical surfaces compared to macaques, offering the possibility of imaging directly from outside the cortex chamber. This releases the constraints on illumination design posed by current chamber insert sizes, allowing our system to achieve an even larger FOV in vivo (Supplementary Fig 7). An additional challenge of imaging marmoset brains is the smaller size of the brain compared to macaques. The curved nature of the brain surface can potentially impact imaging quality, where lens-based systems have limited depth of focus. One distinct advantage of our system compared to current lens-based devices is the capability to perform digital refocusing. By using point spread functions obtained at different depths, this digital refocusing mechanism enables us to bring areas that are out of focus back into focus (as shown in Supplementary Fig 5e). This unique refocusing ability is a key advantage for imaging over curved surfaces like the surface of the marmoset brain. To facilitate use in even smaller animals or the use of multiple microscopes over different brain areas in the same animal, future designs can be made smaller and lighter by using a small and lightweight sensor printed circuit board (PCB) and electronics, as well as lightweight 3D printed materials for the housing. Additionally, incorporating miniaturized OLEDs for on-chip illumination can significantly reduce the form factor of the device and facilitate its integration, potentially leading to the development of an implantable device[86,87].

Overall, in this paper, we have demonstrated that Bio-FlatScopeNHP can capture calcium dynamics in macaque V1 at the columnar scale and provide high-quality orientation map information from a head-unrestrained NHP, opening a path for studying brain activity in these animals under more naturalistic conditions.

## Methods

### Device fabrication

The prototype was constructed utilizing a commercially available board level camera (Imaging Source, DMM 37UX187-ML) with a monochrome Sony CMOS imaging sensor (IMX178LLJ with 6.3 MP and 2.4 μm pixels). The prototype employed a 3.5 mm sensor-to-mask distance and an approximate 3 mm working distance for fixed samples and in vivo imaging.

The PSF pattern for the phase mask was generated by using Canny edge detection on a randomly generated Perlin noise (Supplementary Fig 3) with a feature size of 6 μm, which was selected based on the fabrication limit. The phase mask was then designed using a phase retrieval algorithm[58] using the PSF pattern. The phase mask was fabricated using a 3D maskless two-photon photolithography system (Nanoscribe, Photonic Professional GT) in high-resolution dip-in liquid lithography mode. The mask was fabricated on a 700 μm thick fused silica substrate using a photoresist (IP-Dip). The laser power used for fabrication was 65% of the maximum power, and it should be adjusted based on systems for optimal results by clear visualization in the real-time monitoring software. After exposure, the fabricated mask was immersed in SU-8 developer for 20 min, followed by a 2 min soak in isopropyl alcohol. The size of the fabricated phase mask is 2.3 mm × 2.3 mm and has a height of 1 μm with a 200 nm step and a 1 μm fabrication pixel size. The substrate was laser cut to 7 mm × 7 mm to fit the design of the housing and fitted with an opaque mask to create an aperture containing only the phase mask. A hybrid filter set[49,62] was attached after the phase mask, consisting of a commercially available adsorption filter (Kodak, Wratten 12) and a custom-designed interference filter (Chroma, ET525/50 m) with a thickness of 500 μm. The filters and

phase mask were housed in a 3D printed enclosure (printed using Formlabs Form 3).

The integrated illumination system consisted of four surface-mounted LEDs (LXML-PB01-0030) with excitation filters (Chroma, ET470/40x). The LEDs were symmetrically positioned around the imaging module and angled at 40 degrees to direct light towards the central FOV (Supplementary Fig 7). The excitation power measured at the 3 mm working distance was approximately 0.3 ~ 1.0 mW/mm², which is comparable to that of the table-top widefield system (0.1 ~ 0.2 mW/mm²) and sufficient for one-photon calcium imaging in the macaque cortex. The entire device was mounted on the cortex chamber using a 3D-printed holder (printed using ProJet MJP 2500).

### Device calibration

A one-time calibration procedure must be completed to record the experimental PSF of the mask prior to conducting imaging experiments. The point source we used for calibration was a 10 μm pinhole that was illuminated by a green LED (Thorlabs, M530L4) placed behind an 80-degree holographic diffuser. Images of the central PSFs were obtained at 20 μm intervals across a working distance range of 0.5 mm–6 mm. These central PSFs were used in fast reconstructions using a shift-invariant deconvolution model to assess the positioning of the device quality of the imaging. For the shift-variant deconvolution model, calibration was performed at 20 μm intervals within a working distance range of 2.9 mm–3.1 mm. At a specific depth, the distance between each calibration measurement was 1 mm, and a 9 × 9 grid was calibrated (81 PSFs for one depth) on the imaging plane, covering an imaging area of 8 mm × 8 mm. The calibrations were performed automatically by using programmable motorized linear translation stages (Thorlabs, LNR502). All calibration images were averaged through five captured images to improve the signal-to-noise ratio. Calibration across a range of working distances enables digital refocusing of the images in post processing.

### Image reconstruction

Image reconstruction is a crucial process for lensless microscopes and is commonly approached as a convex optimization problem. To achieve fast reconstruction using single PSF captured at the central FOV, we efficiently solved the following minimization problem with Tikhonov regularization added to the deconvolution to suppress noise amplification:[49]

$$\hat{\mathbf{i}} = \arg\min_{\mathbf{i} > \mathbf{0}} ||\mathbf{b} - p{*}\mathbf{i}||_F^2 + \frac{\gamma}{2}||\mathbf{i}||_F \qquad (1)$$

where $*$ represents convolution, $\hat{\textit{i}}$ is the estimated scene, $\mathbf{i}$ is the actual scene, $\mathbf{b}$ is the measured signal from the sensor, $p$ is the PSF at the depth of the scene, $\gamma$ is the weight of regularization, and $||\cdot||_F$ is the Frobenius norm. To minimize computational complexity, this optimization problem can be solved in closed form using Wiener deconvolution as:

$$\hat{\mathbf{i}} = F^{-1}\left(\frac{F(p)^{*} \odot F(\mathbf{b})}{|F(p)|^2 + \gamma}\right), \qquad (2)$$

where $\odot$ represents the Hadamard product, $(\cdot)^{*}$ is the complex conjugate operator, $F$ represents the Fourier transform, and $F^{-1}$ denotes the inverse Fourier transform. The processing time on an 8-Core Processor (AMD Ryzen 7 3700X, 3.59 GHz) is approximately 0.5 s for one 8-bit frame with 2048 × 2048 pixels using MATLAB. The fast reconstruction speed allows for real-time feedback on the device position and initial image quality check, making it practical for on-site use.

After the imaging experiment, all captured data were reconstructed using spatially variant PSFs obtained at the imaging depth,

resulting in improved reconstruction quality compared to using the shift-invariant model[51]. The reconstruction depth was selected based on the sharpness of features of interest in the region of interest, with the goal of obtaining the best reconstructed results. The scene $\mathbf{i}$ and the background $\mathbf{g}$ were jointly estimated by solving a regularized minimization problem with Tikhonov regularization in the deconvolution:

$$\hat{\mathbf{i}}, \hat{\mathbf{g}} = \arg \min_{\mathbf{i}, \mathbf{g} > \mathbf{0}} ||(\Phi\mathbf{i} + \mathbf{g}) - \mathbf{b}||_2^2 + \frac{\gamma}{2}||\mathbf{i}||_2^2, \qquad (3)$$

where $\Phi$ is the matrix representation of the spatially variant PSFs. The scene $\mathbf{i}$ and the low-frequency background $\mathbf{g}$ were jointly solved by using FISTA[88], with the constraint that the Direct Cosine Transform (DCT) coefficients of $\mathbf{g}$ are set to zero outsize the $5 \times 5$ lowest frequency components. The reconstruction time on a Nvidia GeForce RTX 2070 GPU is approximately 3 min for one 8-bit frame with $2048 \times 2048$ pixels.

## Sample preparation for spiking HEK cells

Spiking HEK 239 cells[64] were cultured in DMEM-F12 (Lonza) supplemented with 10% phosphate-buffered saline (Gibso) and 1% penicillin/streptomycin (Lonza) at 37 °C in a 5% $CO_2$ environment. The glass coverslips, with a diameter of 12 mm, were treated with polydimethylsiloxane to create circles 300 μm in diameter, with a 300 μm spacing between them, using a Lumen 3D printing system[65,89]. The cells were seeded on the coverslips, with 10,000 to 20,000 cells per coverslip, 24−48 h before imaging, resulting in isolated colonies of HEK 293 cells with well-defined size and geometry. The cells were incubated with 2 μM Calcein-AM 30 min prior to imaging. The coverslip was then transferred to a petri dish containing 1 mL PBS for live cell imaging. Both Bio-FlatScopeNHP imaging and ground truth imaging were conducted within 30 min of transferring the coverslip to the petri dish.

## Animal for in vivo imaging experiment

All Experiments were approved by University of Texas Institutional Animal Care and Use Committee (IACUC) under protocol lAUP-2023-00063 and conform to NIH standards. One male rhesus monkey (macaca mulatta, 8 years old) was used in this study. The monkey weighted between 9 and 10 kg.

## Surgical procedure

All procedures have been approved by the University of Texas Institutional Animal Care and Use Committee and conform to NIH standards. Our general experimental procedures in behaving macaque monkeys have been described in detail previously[12,13,47]. Briefly, the animal was implanted with a metal head post and a metal recording chamber located over the dorsal portion of V1, a region representing the lower contra-lateral visual field at eccentricities of 2−5 deg. Craniotomy and durotomy were performed in order to obtain a chronic cranial window. A transparent artificial dura made of silicone was used to protect the brain while allowing optical access for imaging.

Details of the method for injecting virus in the animal have been described previously[12,13,47]. The virtual we used is AAV1-CaMKIIa-NES-GCaMP6f with a titer of 4.8E12 viral genomes per microliter, which only infected the excitatory cells. In summary, a glass pipette (20 μm tip diameter) was first lowered through an opening in the imaging chamber, puncturing the pia. Injections were made at depths of 1.5, 1.0, and 0.5 mm using a Nanoject II. At each depth, 0.5 μL was delivered manually in $10 \times 50$ nL steps, with approximately 30 s pauses between steps. The injection was performed in a hexagon array with a neighbor distance of 2 mm across an area of 50−100 mm², resulting in a consistent and uniform expression. In our previous study[13], we showed that the viral expression was relatively uniform within the imaging ROI using this injection strategy. The effect of slight nonuniformity

expression will only present in a low spatial frequency, and can be filtered out during the signal processing using a bandpass spatial filter (0.8 cyc/mm cutoff). The quantitative measurements and computational model from our previous study[13] suggest that the measured signals reflect the pooled spiking activity of layer 2/3 neurons rather than their pooled synaptic potentials or pre-synaptic inputs.

## Recording sessions for in vivo imaging of behaving animals

The experimental techniques for optical imaging in behaving monkeys have been described in detail elsewhere[12,13,90,91]. We imaged GCaMP signals from V1 of one monkey with a well-shape insert with 8 mm height, whose bottom is 200 μm-thick glass with 16 mm diameter. The insert has an inner diam of 21 mm and an outer diam of 23 mm. The imaging chamber has an outer diameter of 36 mm and an inner diameter of the opening 26 mm. To limit the imaging FOV, a square aperture was constructed from double coated carbon conductive tape (TED PELLA, INC) and placed at the bottom of the chamber insert. For position tuning imaging, the aperture size was 5 mm × 5 mm, and for orientation imaging the aperture size was 4.5 mm × 4.5 mm. The ground truth epi-fluorescence imaging was performed with a custom designed imaging system based on an sCMOS camera (PCO4.2 camera) using the following filter sets: GCaMP, excitation 470/24 nm, dichroic 505 nm, emission 515 nm cutoff glass filter. Illumination was obtained with an LED light source (X-Cite 110 LED). The Bio-FlatScopeNHP imaging was performed in the same recording session, and illumination was obtained by using the integrated light source on the device. The Bio-FlatScopeNHP was mounted on top of the implant by using two 8−32 screws embedded in dental acrylic (Supplementary Fig 5c, d). Data acquisition was time locked to the animal's heartbeat. Slow hemodynamic signals usually start 2 s after stimulus onset and peaks at 4−6 s[4,6,92], which is much longer than the stimulus and imaging duration used in this study. Within the short stimulus time, the largest artifact is the heartbeats (at 2−3 Hz)[93], and this artifact can be reduced by synchronizing the data acquisition to the electrocardiogram (Supplementary Fig. 15). This allowed us to remove the average blank time course from all trials, which significantly reduced the fast hemodynamic artifacts. The imaging was performed under the same conditions and within a two-hour time window. Both devices were recorded at 20 Hz for GCaMP imaging. For both position and orientation tuning experiments, visual stimulus conditions were randomly interleaved, repeated 10 times each, and mixed with blank fixation trials.

## Imaging stability on head-unrestrained animals

We have evaluated the stability of the images captured using the Bio-FlatScopeNHP during head-unrestrained recording. We captured an image before releasing the head restriction of the animal, and this image was used as the template for motion correction. We analyzed the motion in 10 trials in one head-unrestrained experiment session. The absolute maximum displacements in the medial-lateral and anterior-posterior direction were 17.2 μm and 27.2 μm, respectively (Supplementary Fig 13). These small displacements caused by motion could be digitally corrected by registering each frame using vascular structures.

## Data analysis for imaging position tuning

The process for analyzing the ground truth signals captured by the widefield microscope involved several steps. First, the average time course in blank trials was subtracted from the response in each stimulus condition. Next, the mean residual response in the 200 ms period before response onset was removed to reduce the effect of sources of noise like the heartbeat artifact and other slow, widespread fluctuations in the signals. The imaging signals were then averaged across repeats. To compute the spatial tuning curves, six regions of interest (ROIs) were selected, each with a size of 0.3 × 0.3 mm² and spaced approximately 0.37 mm apart (as shown in Fig. 3d, e). The average response from the ROIs was fitted with a 1D Gaussian function.

The Bio-FlatScopeNHP images were reconstructed frame by frame from the raw captured videos using spatially variant PSFs, and downsized to the same size as images captured by the ground truth microscope. The analysis process was identical to that carried out on the widefield microscope images. The data analysis was conducted using Matlab.

## Data analysis for imaging orientation tuning

The Bio-FlatScopeNHP images were obtained by reconstructing each frame of the raw captured videos using spatially variant PSFs, followed by downsizing them to the same size as images captured by the ground truth microscope. The remaining data analysis process was identical for both types of images, and was conducted using Matlab. Initially, the blank trials were subtracted, and the mean residual response was removed, after which the data was averaged, and this processing was the same as that used for processing position tuning data. The ROI was selected based on the RMS map of the response, and the areas with the strongest 4 Hz signals were identified by choosing the overlapping regions between the ground truth and Bio-FlatScopeNHP captures, where the RMS was greater than one-third of the maximum value (Supplementary Fig 9). Once the ROIs were selected, we obtained the orientation response by calculating the first harmonic amplitude using Fast Fourier Transform (FFT) of the temporal response. To study the spatial scale of the orientation columns, we selected a range of spatial frequencies based on the average 1D amplitude spectrum[66] (Figs. 4f, 5d). The peak at approximately 1.2 cycles/mm corresponds to the periodicity of the orientation columns, with an average cycle of around 0.83 mm. To remove any non-orientation-selective responses and high frequency noise, we applied a bandpass spatial filter of 0.8–2.5 cycle/mm. This spatial filtration removes components with periods larger than 1.25 mm and smaller than 0.4 mm.

The mean and standard deviation of the response image at each pixel location were then calculated separately for each condition in each session[94]. The d' map was calculated by using the mean-response image of 0 degree $m_0(x,y)$ and 90 degree $m_{90}(x,y)$ and SD-response image of 0 degree $\sigma_0(x,y)$ and 90 degree $\sigma_{90}(x,y)$:

$$d'_{map}(x,y) = \frac{m_0(x,y) - m_{90}(x,y)}{\sqrt{(\sigma_0^2 + \sigma_{90}^2)/2}} \tag{4}$$

The decision variables $DV$ (Fig. 4e) were then calculated by using the weights equal to the d' maps:

$$DV_{0_i} = \sum_{x,y} m_{0_i}(x,y) d'_{map}(x,y), \tag{5}$$

$$DV_{0_i} = \sum_{x,y} m_{0_i}(x,y) d'_{map}(x,y), \tag{6}$$

The discriminability between 0 degree and 90 degree can be calculated as:

$$d'_{DV} = \frac{m_{DV0} - m_{DV90}}{\sqrt{(\sigma_{DV0}^2 + \sigma_{DV90}^2)/2}}, \tag{7}$$

where $m_{DV0}$ and $m_{DV90}$ are the average of 0 degree and 90 degree decision variables, $\sigma_{DV0}$ and $\sigma_{DV90}$ are the standard deviations of 0 degree and 90 degree decision variables.

To verify that the resulting maps reflected the orientation columns, we calculated the correlation between pairs of maps as a function of their stimulus orientation differences (Figs. 4g & 5e). Next, we used a vector summation method[66] to obtain the complete orientation map from single orientation response maps (Figs. 4h & 5f). In these maps, color represents the preferred orientation at each location, and the saturation indicates the strength of orientation tuning.

## Statistics and Reproducibility

For Fig. 2c, d, the imaging of the USAF test target and Convallaria with both the Bio-FlatScopeNHP and 4x objective were repeated more than 10 times, with similar results. For Fig. 2e, f, the imaging of patterned live spiking HEK293 cells was performed 3 times, with similar results. For Fig. 3f, the position tuning experiments were performed 2 times on 2 different days on a head-fixed male macaque, with similar results. For Fig. 4c, h, the orientation tuning experiments were performed 2 times on 2 different days on a head-fixed male macaque, with similar results. For Fig. 5c, f, the orientation tuning experiments were performed 2 times on the same day on a head-unrestrained male macaque, with similar results. Additional results were presented in Supplementary Information.

## Reporting summary

Further information on research design is available in the Nature Portfolio Reporting Summary linked to this article.

## Data availability

The main data supporting the results of this study are available within the paper and its Supplementary information. Bio-FlatScopeNHP CAD design, phase mask design, all costume codes and sample data are provided on a Github repository[95]. The raw and analyzed datasets generated during the study are too large to be publicly shared, but they are available for research purposes from the corresponding author upon request. Source data are provided with this paper.

## Code availability

Custom MATLAB codes are available on a GitHub repository[95] (https://github.com/JiminWu/Bio-FlatScopeNHP).

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

## Acknowledgements

This work was supported in part by DARPA grant N66001-17-C-4012 for JTR AV & ES, NIH grant RF1NS110501 for J.T.R. & A.V., NSF grant IIS-1730574 and IIS-1652633 for A.V., NIH grants R01EY016454 and U19NS118284 for E.S.; This research is partially sponsored by the Defense Advanced Research Projects Agency (DARPA) through Cooperative Agreement D20AC00002 (for J.T.R. & A.V.) awarded by the U.S. Department of the Interior (DOI), Interior Business Center. The content of the information does not necessarily reflect the position or the policy of the Government, and no official endorsement should be inferred. We thank G. Duret for preparing the Spiking HEK cells samples; V. Boominathan for helpful discussions and the scanned electron micrograph of the phase mask; and D. Yan for the help on calibration setups.

## Author contributions

J.W. designed, fabricated, and characterized the Bio-FlatScopeNHP prototype, and developed the reconstruction model. Y.C. designed and constructed the table-top wide-field imaging system, performed virus injections, and performed animal training. Y.C. and E.S. designed the in vivo experiments with visual stimulus. J.W. and Y.C. performed the in vivo imaging experiments and data analysis. A.V., E.S., and J.T.R. provided guidance and assistance with all aspects of the work. All authors contributed to the writing of the manuscript.

## Competing interests

J.T.R. is cofounder of, holds equity in, and receives payment from Motif Neurotech. The remaining authors declare no competing interests.
