## [Peer Review File NEW · Nature Communications]

Mesosopic calcium imaging in a head-unrestrained male non-human primate using a lensless microscopeEditorial Note: This manuscript has been previously reviewed at another journal that is not operating a transparent peer review scheme. This document only contains reviewer comments and rebuttal letters for versions considered at *Nature Communications*.

Reviewer #1 (Remarks to the Author):

This is a follow up of from a reviewing thread started in Nature Methods. I will therefore not repeat earlier general comments about the manuscript.

Thanks for adding qualifiers to the text, particularly in the Discussion, to leave readers with a realistic assessment of the capabilities of this microscope in its current state of development. The overall concept of the microscope is promising, with considerable potential and the text now makes that point well. I have no further comments.

I look forward to further development of the microscope, e.g with an additional, interdigitated illumination / absorption pathway to help with hemodynamic correction, for extended imaging trials and more realistic experimental setups. Particularly in marmosets.

(signed) Aniruddha Das

Reviewer #2 (Remarks to the Author):

My previous comments reflected, in part, my lack of being convinced that the work as it stands would be sufficiently impactful to a broad audience, and this concern remains unabated for consideration at Nature Communications.

As it stands, the authors have not made any changes in the manuscript (other than additions to the discussion) in response to my concerns, and I don't see how I can reasonably change my view of the work's impact. Notably, the issues regarding hemodynamic correction are still unaddressed (and were also brought up by reviewer 1).

1. Whether there is any substantial hemodynamic contamination to the visual stimulus responses in the present paper is somewhat beside the point. I note that their claim this is not an issue for the early response periods studies here, after removal of heartbeat-locked artifacts is not terribly convincing. The authors are trying to demonstrate a highly impactful new method. They are thus obligated to show that the system can and will work under broadly applicable circumstances. For example, can this system collect accurate spontaneous activity data sets that are not time-locked to a stimulus? Can this system collect usable data outside the "1 second" window for sensory-evoked responses? Beyond these questions, the authors don't even provide any raw data showing that there is a heartbeat artifact and that it is successfully removed with their method.

2. It feels that most of the push-back originates from these data coming from a single animal and the authors are unable or unwilling to do additional experiments to address reviewer concerns. I appreciate the challenges of primate work, but this is simply not acceptable for a study claiming broad impact.

3. If the biological data (visual responses) are ancillary and simply to show that the camera can collect data, the manuscript ultimately rests on the technical development of a new mesoscopic camera system for primates. I am unconvinced this point alone will be of widespread interest in the absence of demonstrated biological impact. I feel strongly that the authors must present at least some biological findings that (1) are made uniquely possible by this new camera and (2) are rigorously obtained using the current standards in the widefield imaging field.

We thank the editor and the reviewers for their valuable feedback, which we believe has helped to improve the manuscript. We provide a point-by-point response to reviewers' comments below.

Reviewer #1

This is a follow up of from a reviewing thread started in Nature Methods. I will therefore not repeat earlier general comments about the manuscript.

Thanks for adding qualifiers to the text, particularly in the Discussion, to leave readers with a realistic assessment of the capabilities of this microscope in its current state of development. The overall concept of the microscope is promising, with considerable potential and the text now makes that point well. I have no further comments.

I look forward to further development of the microscope, e.g with an additional, interdigitated illumination / absorption pathway to help with hemodynamic correction, for extended imaging trials and more realistic experimental setups. Particularly in marmosets.

We thank the reviewer for the positive assessment and for recognizing the potential of our work. We are especially thankful for the invaluable suggestions and comments from the reviewer throughout the revision process, which significantly help us improve our manuscript.

Reviewer #2

My previous comments reflected, in part, my lack of being convinced that the work as it stands would be sufficiently impactful to a broad audience, and this concern remains unabated for consideration at Nature Communications.

As it stands, the authors have not made any changes in the manuscript (other than additions to the discussion) in response to my concerns, and I don't see how I can reasonably change my view of the work's impact. Notably, the issues regarding hemodynamic correction are still unaddressed (and were also brought up by reviewer 1).

1. Whether there is any substantial hemodynamic contamination to the visual stimulus responses in the present paper is somewhat beside the point. I note that their claim this is not an issue for the early response periods studies here, after removal of heartbeat-locked artifacts is not terribly convincing. The authors are trying to demonstrate a highly impactful new method. They are thus obligated to show that the system can and will work under broadly applicable circumstances. For example, can this system collect accurate spontaneous activity data sets that are not time-locked to a stimulus? Can this system collect usable data outside the "1 second" window for sensory-evoked responses? Beyond these questions, the authors don't

even provide any raw data showing that there is a heartbeat artifact and that it is successfully removed with their method.

This paper serves as a proof-of-principle demonstrating that the lensless microscope can achieve imaging quality comparable to that of a standard table-top widefield microscope. Moreover, we show the capability of this microscope to extract small neurological signals, such as orientation columns maps from a head-unrestrained animal. To demonstrate the imaging quality of the system, we use the established experimental settings employed in previous work (e.g., Chen, Y. et al. Nature Neuroscience 2006, Seidemann, E. et al. eLife 2016), incorporating a heartbeat triggered visual stimulus.

Future work can certainly build on this proof-of-concept to develop a more broadly applicable method that could facilitate continuous recording by incorporating an additional imaging path designed to record the hemodynamic signals. This is outside the scope of the current work and involves a change in filters and illumination, while the fundamental concept of lensless microscopy would remain the same. We anticipate that the system can accurately collect spontaneous activity data outside the 1 second time window and without heartbeat triggered stimulus when we simultaneously capture the hemodynamic signals, but this will be the topic of future work. Thanks to the reviewer's previous review we made the changes in the discussion to emphasize this point.

Removing heartbeat artifact using synced stimulus is widely used and it was first documented in (Grinvald et al., Journal of Neuroscience, 1994). To show the heartbeat artifact removal we added a supplementary figure (Supplementary Figure 15). This shows the raw time traces captured by the table-top widefield microscope, and a comparison before and after the heartbeat artifact correction. Heartbeat artifact is clearly shown in Supplementary Figure 15b, and got removed after the subtraction of averaged blank trials. It's important to highlight that despite the presence of a heartbeat artifact, we still captured a robust 4 Hz stimulus signal, just with a reduced signal-to-noise ratio (SNR).

Supplementary Fig. 15 | Heartbeat artifact removal. a, Average time course of blank trials over the center 2 mm \times 2 mm area of the FOV. The signal fluctuation is caused by the heartbeat. Shaded area \pm SEM. b, Average time course of GCaMP response to the flashed gratings (60 degree stimulus) over the

center 2 mm × 2 mm area of the FOV before heartbeat artifact correction. Shaded area ± SEM. c, Average time course of GCaMP response to the flashed gratings (60 degree stimulus) over the center 2 mm × 2 mm area of the FOV after heartbeat artifact correction. Shaded area ± SEM. Data presented here were captured using the table-top widefield microscope. All traces represent average across 10 repeats. Source data are provided as a Source Data file.

2. It feels that most of the push-back originates from these data coming from a single animal and the authors are unable or unwilling to do additional experiments to address reviewer concerns. I appreciate the challenges of primate work, but this is simply not acceptable for a study claiming broad impact.

This paper primarily focused on the technical development of the new lensless microscope. Using this new device, we performed multiple experiments on a macaque over various days spanning a 6-month duration. Specifically, we performed position tuning experiments and orientation tuning experiments more than twice each, consistently obtaining stable results (additional data in Supplementary Information). These consistent outcomes clearly demonstrate the stability of our system. Additional animals and experiments are unlikely to provide any new insights or information.

3. If the biological data (visual responses) are ancillary and simply to show that the camera can collect data, the manuscript ultimately rests on the technical development of a new mesoscopic camera system for primates. I am unconvinced this point alone will be of widespread interest in the absence of demonstrated biological impact. I feel strongly that the authors must present at least some biological findings that (1) are made uniquely possible by this new camera and (2) are rigorously obtained using the current standards in the widefield imaging field.

This paper focuses on the development of a miniaturized lensless microscope for non-human primates, capable of achieving similar imaging quality to standard table-top widefield microscopes. Using this microscope, we successfully imaged the first orientation columns map from a head-unrestrained macaque - uniquely made possible by the Bio-FlatScopeNHP. There was no guarantee that these maps would match the head-fixed maps, which is a finding in itself. With the future improvement incorporating hemodynamic correction, this system can assist researchers in observing neurological signals under more naturalistic conditions. This would allow researchers to study the impact of natural head and eye movements on sensory areas like V1 and the auditory cortex, as well as sensory-motor regions such as the premotor cortex and frontal eye fields.